# What Is Wrong with Eating Pets? Wittgensteinian Animal Ethics and Its Need for Empirical Data

**DOI:** 10.3390/ani13172747

**Published:** 2023-08-29

**Authors:** Erich Linder, Herwig Grimm

**Affiliations:** 1Unit of Ethics and Human-Animal-Studies, Messerli Research Institute, University of Veterinary Medicine Vienna, Medical University of Vienna, University of Vienna, 1210 Vienna, Austria; herwig.grimm@vetmeduni.ac.at; 2Vienna Doctoral School of Philosophy, University of Vienna, 1010 Vienna, Austria

**Keywords:** Wittgensteinian ethics, animal ethics, empirical data, empirical ethics

## Abstract

**Simple Summary:**

Wittgensteinian ethicists argue that we should not rely on a set of principles if we want to know how to treat non-human animals. Instead, we should look at how we witness and encounter animals in our lives. We admire wild animals, we feed our pets, and we cure them as patients. For Wittgensteinian animal ethicists, moral reflection should start from these ways of thinking about animals. However, our understanding of animals can change depending on context and circumstance. Not everyone thinks about animals in the same way. It is, therefore, important that Wittgensteinian animal ethicists are informed about the ways that people think about animals. We argue that this information should come from data gathered by social sciences such as sociology, psychology or anthropology.

**Abstract:**

Wittgensteinian approaches to animal ethics highlight the significance of practical concepts like ‘pet’, ‘patient’, or ‘companion’ in shaping our understanding of how we should treat non-human animals. For Wittgensteinian animal ethicists, moral principles alone cannot ground moral judgments about our treatment of animals. Instead, moral reflection must begin with acknowledging the practical relations that tie us to animals. Morality emerges within practical contexts. Context-dependent conceptualisations form our moral outlook. In this paper, we argue that Wittgensteinians should, for methodological reasons, pay more attention to empirical data from the social sciences such as sociology, psychology or anthropology. Such data can ground Wittgensteinians’ moral inquiry and thereby render their topical views more dialectically robust.

## 1. Introduction

Since the so-called ‘empirical turn’ in bioethics [1,2,3,4], ethicists have increasingly appreciated the value of empirical data from the social sciences. Such data can provide guidance in dealing with applied ethical problems. It has become apparent that universal moral principles alone cannot adequately accommodate the complexities of concrete, practical situations [5]. Discussing ethical solutions from a purely theoretical standpoint is not enough. If we want to provide meaningful criticism and advice, we must engage directly with the attitudes, values, and complexities inherent in each ethical scenario.

How exactly can empirical data from the social sciences contribute to ethical reflection? Bioethicists have proffered a variety of answers to this question. Some argue that ethical judgments should be guided by principles; empirical data can only help contextually refining these judgments [6] (pp. 25–26). Others deny that principles are helpful (e.g., [7]) and place more emphasis on contextual and empirical factors (for an overview, see [5]). As the debate has developed, intermediate positions have emerged in which empirical data inform ethical reflection to varying degrees (for a systematic review, see [8]).

Since there is no consensus on which of the above stances is correct, we think that it is, at least, important for applied ethicists to make their position explicit by (1) disclosing their theoretical assumptions and (2) clarifying which sort of connection their view has to empirical data from the social sciences. Should empirical data merely support ethical judgments by refining principles or should it be the very ground of moral judgments? Indeed, if one rejects the idea that universal principles ground moral judgments, then right and wrong must be determined relative to context. And, contexts are properly explored through empirical inquiry, that is, through situated investigations that generate useful data. For example, by conducting focus groups or qualitative interviews.

Animal ethics has witnessed a shift towards greater use of empirical data (this special issue is exemplary). Nonetheless, the debate has traditionally been dominated by so-called orthodox approaches [9,10]. What is distinctive of advocates of orthodox approaches—be they deontologists [11,12] or utilitarians [13,14,15]—is that they employ moral principles established a priori that serve as a point of reference for moral evaluations. Even if some advocates of orthodox approaches increasingly consider empirical data from the social sciences, they still discern what is right or wrong from moral principles. Orthodox animal ethicists make use of empirical data hold that what the folk think is the right way to treat animals is only relevant as a means to implement the right course of action (e.g., by nudging them in the right direction [16] (pp. 176–181)). Some advocates of orthodox approaches might suggest that, given our psychological dispositions towards animals, we should favour a particular moral theory, one that accommodates our tendency to favour humans [17]. Nonetheless, the relevant mindset remains: we first identify the correct theory (with associated principles) and only then do we apply it in the most effective way. As we will see, the connection between empirical data and animal ethics has, to date, only taken into consideration the orthodox position. Such is, for example, the stance taken by Kirsten Persson and David Shaw in their ground-breaking article “Empirical Methods in Animal Ethics” where they systematise how orthodox accounts would benefit from empirical data from both the natural and social sciences [18].

However, orthodox approaches had their critics: Wittgensteinian ethicists. The latter criticised the former for being reductionistic and rationalistic [9]. Wittgensteinians emphasise the inextricable link between morality and our form of life; the latter being “the intertwining of culture, worldview, and language” [19] (p. 124). They argue that any moral problem is best understood by considering the context within which it emerges. Every morally relevant concept and every morally relevant action has its origin in the network of language games and practices that constitute our forms of life. Morality cannot be understood from an external point of view nor can it be reduced to a set of moral principles established a priori as e.g., utilitarian claim to do cf. ([13]). Rather, the meaning of each moral concept should be understood through its use in different practices, in different language games.

Like so-called anti-theorists [20] (pp. 21–22), Wittgensteinians argue that a moral judgment about some situation “cannot be coded in explicit and general principles” [20] (p. 20). Rather, it “involve[s] a complicated understanding of the good life” [20] (p. 20). And such an understanding is always situated within a context, it is formed through the plurality of concepts that constitute our forms of life. The promise that our moral judgments might be enriched beyond classical uses of deontological and consequentialist principles has prompted something of a revolution in ethical studies in the wake of Wittgenstein, e.g., [21,22] and an interest towards applied problems in ethics in general, e.g., [23] and animal ethics in particular, e.g., [24,25,26,27,28]. However, in animal ethics, little has been said about the relationship between Wittgensteinian approaches and social sciences (Hannah Winther’s recent work [3]—inspired by the philosophy of Cora Diamond—is an exception).

Note that our aim in this paper is not to settle the question of whether Wittgensteinian approaches indeed provide a better understanding of moral problems than orthodox approaches. There is ongoing debate in this regard. We aim, rather, to show that, given their peculiar approach, Wittgensteinians should make extensive use of empirical data. Indeed, the fact that Wittgensteinians ground moral judgments in our forms of life (rather than in principles) raises the question of how they should interpret data from the social sciences. When sociologists, anthropologists, and psychologists describe how a community perceives animals, for example, they are describing the concepts that constitute their form of life. They can, for example, show how animals are conceptualised and what sort of morally relevant relationships with them are de facto present. If so, then it seems natural to draw from the social sciences when engaging in ethical reflections and discussions. However, in animal ethics, Wittgensteinians have so far only explored the concepts we live by through discussions of the work of poets and novelists [25,27,28,29,30,31,32] or by drawing on anecdotal experiences and on their own preconceptions of what an animal is (cf. [27,30]). We contend that this is not sufficient. Hence, our primary aim here is to argue that Wittgensteinians should, for methodological reasons, rely on the social sciences if they wish to present a robust alternative to the orthodox approach. That is, they should interpret empirical data about the specific contexts and situations they are interested in as the starting point for any ethical reflection.

In Section 2 and Section 3 of this paper, We will make explicit the Wittgensteinians’ peculiar approach to ethics by exploring how the orthodox and the Wittgensteinian approaches substantiate moral judgments. Wittgensteinians, as opposed to orthodox approaches, consider different sorts of reasons to be relevant when justifying moral judgements. For example, the fact that an animal is a pet gives us *already* a reason to act in a certain way (without the need to invoke any principles). The fact that we *think* about some animals in this way is taken to be a reason for why it is wrong to eat them. Asking whether it is wrong to eat them is seen as an ill-guided question. This will show how Wittgensteinians consider descriptions of how we de facto already think as bearing a normative potential.

In Section 4 and Section 5, we argue that social sciences can inform us with relevant descriptions about how we already think about moral issues and conclude that a systematic inquiry into how people actually think about animals can provide a more substantial foundation for Wittgensteinian ethical reflection than does the anecdotal approach they have used so far. Wittgensteinian reflections should not shy away from engaging in interdisciplinary work with social scientists. Indeed, the former’s perspective provides a profound insight in the work of social scientists: the data they collect is normatively relevant.

In Section 6, we briefly address some possible worries such as the concern that Wittgensteinian approaches lead to relativism. Even if our paper is not aimed at defending a Wittgensteinian position we think it is important not to trivialise their approach as it could indeed provide a more context-sensitive approach to animal ethics.

## 2. What’s Wrong with Eating Pets? Orthodox versus Wittgensteinian Answers

In this section, we take a first step towards defending our claim that Wittgensteinian animal ethicists [24,25,26,27,28,29,30,31,32] should rely on empirical data from the social sciences to develop a substantial foundation for their ethical reflections. This step is grounded in the idea—held by the Wittgensteinians and reflected in the way they support their judgments—that descriptions of how we already think about animals carry a normative potential. To better elucidate this idea, we will contrast the way in which Wittgensteinians and utilitarians—one of the dominant positions in orthodox animal ethics—ground ethical judgments.

Consider the following question: “What’s wrong with eating pets?”.

A utilitarian e.g., [13,14,15] might respond:

“Eating pets is wrong because one must kill the pet to eat it and this would cause, on balance, more harm than good in the world” (In this example we are assuming that killing the animal would bring about the highest amount of harm. Hence, e.g., we assume that (1) the harm inflicted to the animal would not outweigh the pleasure derived from eating it and that (2) there are alternatives to eating an animal that would bring about better consequences).

It is important to note that utilitarians mainly give two reasons for why eating pets is wrong:We should adhere to the principle of maximising overall happiness (or pleasure or well-being).Non-human animals are sentient beings.

These reasons are aimed at supporting the claim that eating pets is wrong. They are aimed at justifying the utilitarian’s above answer: the utilitarian is justified in holding that it is wrong to eat a pet because they can give us these two reasons. Since these reasons justify the morality of a judgment we will call them justificatory reasons.

Note that two different claims are being made in the above case.

Reason 1 involves a normative claim. It concerns how we ought to act: maximise happiness.Reason 2 involves a descriptive claim. The fact that a pet is a sentient being describes how the world is. It tells us a putative fact.

The fact expressed in 2 is normatively inert, provided we do not have a principle instructing our behaviour. This principle need not be the utilitarian one which we have used as an example. It might be a deontological principle that prohibits treating animals as means to an end [11,12]. In any event, the point is that, once we know that the animal is sentient, we must still look for a principle telling us what the right (or wrong) thing to do would be. We do not have a moral reason to refrain from eating the pet if there is no principle that informs us about the right way to act. Once we accept that we should maximise happiness, we can accept that the fact of sentience is a reason not to harm the pet. Being sentient is, then, a reason that justifies our action in a morally relevant way. But, this is only because we hold to a prior moral principle. The fact that we are dealing with a pet does not provide us with any justificatory reason over and above the fact of sentience. What is morally relevant is that we are dealing with a sentient being. Whether it is a pet, vermin, or a farmed animal is per se irrelevant (It is worth noticing that, within a specific context a utilitarian might say that hurting a pet causes an indirect harm to the owner, whereas hurting a pest would probably maximise the happiness among people. So, overall, more happiness would come from hurting the pest.). In utilitarianism as in other orthodox approaches, the animal that we encounter in particular situations is, in a sense, reduced to an impoverished entity detached from any practical context, be it a “being with interest” [13], a “subject-of-a-life” [11], “kin” [33], or a “fellow creature” [34]. In these approaches the various authors seem to look for the animal per se. That is, the essence of the animal that we encounter in every context and to which moral principles hold us responsible. And the way to identify what that animal is, typically, consists in a scientific inquiry about its sentience or other cognitive capacities which, in turn, allow us to determine whether it has the relevant intrinsic properties to be included in the moral community [35].

Wittgensteinians might respond to the question, “What’s wrong with eating pets?” by remarking:

“What’s wrong with you? A pet is not something to eat!”

For the Wittgensteinian, it is crucial that we are dealing with a pet and not merely with a generic sentient being. The Wittgensteinian argues that we should not eat our pets because they are our pets. Saying that they are our pets is already a justificatory reason as it points out at the shared understanding of what ‘pet’ means i.e., how some use such term in a form of life. For the Wittgensteinian, reminding that the animal is a pet should be sufficient, and a demand for further reason is, to some extent, an inappropriate question. However, it is not clear how describing an animal in a certain way counts as a justification. So, on what principle does the Wittgensteinian ground ‘being a pet’ as a justificatory reason?

An advocate of the orthodox approach would claim that all the Wittgensteinian is doing is explaining how the world already is. The Wittgensteinian is saying something like “we humans have this sort of practice that we call ‘pet-keeping’ where we take care of animals and do not eat them”. The orthodox ethicist might argue that the Wittgensteinian is merely providing an explanatory reason, that is, a reason that explains why people do not eat their pets. People do not eat their pets because what ‘being a pet’ means for them entails not eating the pet. People have a form of life that makes it abhorrent to eat their pets. But, the orthodox ethicist might argue, this leaves open the question of whether people really should not eat their pets. Wittgensteinians are simply pointing at how the world (descriptively) is, but they have no (normative) principle that instructs them how to act [36] (p. 376). In other words, the orthodox ethicist is asking for further reasons.

However, the need for further reasons and the appeal to universalizable principles is precisely what Wittgensteinians dispute. They refuse to reduce our moral life to a set of principles that everybody has to abide by. As we will see, for Wittgensteinians, the wrongness that stems from eating a pet cannot be reduced to the violation of a principle and to the fact that a pet is a sentient being. What ultimately makes a reason a justificatory reason is a particular concept we live by within a particular form of life. For Wittgensteinians, moral judgment should not be limited to a “shallow” and “moralistic tone” [27] (pp. 468–469) which is conveyed through principles. It should, instead, allow us to show aspects of a particular form of life: that is, it should take into account the nuances that differentiate the variety of concepts through which agents make sense of the world.

The concept of a ‘pet’ cannot be disentangled into separate evaluative and descriptive components; rather, it carries a particular meaning that those engaged in the practice of pet-keeping can aptly grasp. Thus, questioning why it is wrong to eat a pet means to address questions that don’t need to be asked cf. [37] (p. 56) [38] as they lead us to engage what Bernard Williams calls “one thought too many” [39] (p. 19). Diamond, for example, claims that with these questions one “runs a risk of leaving those fundamental features of our relationship to [pets] which are involved in our not eating them” (In the original text Diamond makes this point about human beings, but the same considerations apply to pets) [27] (p. 467). The mere acknowledgment that we are dealing with a pet already encompasses an understanding of how we should treat them. One might dispute that we are dealing with a pet, but the very fact that one acknowledges that we are dealing with a pet comes with an understanding of how we treat pets.

The same line of criticism applies as well to deontological approaches, even though they seem closer to Wittgensteinians than utilitarians. Indeed, deontologists can formulate obligations specific to the fact that the animal is a pet. Claire Palmer argues, for instance, that we have obligations towards domesticated animals which differ from those towards wild animals due to the particular relationships we establish with the former [40]. Along these lines, one could argue that we also have specific obligations towards pets and therefore claim that also deontologist make reference to concepts such as ‘pet’. The primary difference with Wittgensteinian approaches, as will be clarified in the following sections, lies in their distinct meta-ethical approach. As pointed out by Alice Crary and Lori Gruen, deontologists still adhere to a strict distinction between facts and values, deriving our normativity from universalizable principles [41] (pp. 97–98). Moreover, any obligation we have towards pets will depend on previously established principles—such as the “respect principle”—which apply universally to all beings in virtue of the fact that they are e.g., ‘subjects-of-a-life’ [11]. Therefore, if we have special obligations towards a pet, they will still be grounded on universalizable principles and on a conception of the animal as an animal per se.

In other words, the reasons why we should not harm a pet are established in both deontological and utilitarian accounts by referring to a principle whose violation determines the wrongness of an action, and, ultimately, to the fact that the being in question is sentient. However, for Wittgensteinians, this is not the case. The wrongness of eating a pet can only be understood if one is raised in a community where the practice of pet-keeping holds particular significance.

We will argue that this peculiar understanding of ethics means that Wittgensteinians should employ empirical data as a ground for moral argumentation.

We have explicated how the orthodox approach grounds justificatory reasons in principles and how Wittgensteinians ground them in people’s forms of life. We now flesh out Wittgensteinian ethics to show how simply being a pet can serve as a justificatory reason. That is, how they derive normative claims from a description of how people already think.

## 3. Morality in a Form of Life: Anti-Reductionism and Description in Wittgensteinian Moral Thought

For Wittgensteinians, the wrongness of eating a pet, understood through the concept ‘pet’ and the practice of ‘pet keeping’ cannot be reduced to a prohibition on harming sentient beings to maximise happiness. Instead, these notions demonstrate meaningful ways in which we can understand our relationships with animals.

There are two aspects of the Wittgensteinian approach that highlight how normativity stems from our forms of life.

Firstly, Wittgensteinian ethical inquiry is anti-reductionist. Such anti-reductionism can be understood in two ways:Our understanding of morality is not reduced to a set of principles.The animals we encounter within our form of life—such as “pets”, “patients”, and “companions”—are not reduced to beings with morally relevant properties. Wittgensteinians do not reduce pets to animals per se (the sum of morally relevant properties).

Secondly, Wittgensteinian ethical inquiry is a descriptive approach to ethics. There is no overarching and universal principle, criterion, or theory that can ground moral judgments and provide ultimate solutions. There is rather a plurality of morally relevant concepts that the ethicist should highlight and describe.

Wittgensteinians’ anti-reductionism and descriptivism are grounded in the idea that morality is always constituted within a specific form of life. That is, within the “intertwining of culture, world-view and language” [19] (p. 124) that shape our understanding of the world. Indeed, what we come to understand as moral is always shaped within a particular context through language games. This means that the concepts that we understand as moral gain their meaning through the use of terms within particular contexts and situations. Any morally sound judgment can only be established within the boundaries of practical and linguistic activities. Understanding what is wrong with eating pets must come with an understanding of what it means to live a specific form of life where pets are involved. Paraphrasing Rorty, Diamond states that

[m]oral concern is something we have not as rational beings or as animals with certain capacities but as members of communities within which this or that language of moral deliberation has taken shape [31] (p. 39).

Our understanding of morality—as constituted in a form of life—becomes the starting point for comprehending what is morally justified or unjustified. We can only understand how moral judgments are justified by considering how meaning is constructed within a community. People living in different communities come to understand the world differently as they engage in distinct practices and language usage. There is a “close relationship between the people and their environment and the delicate adjustment of their lives to that environment has come to be encoded into other grammar of their language” [42] (p. 36) cf. [43] (p. 487). Indeed, this is reflected in how animals are classified and described cf. [44,45,46]. The way a Scottish fisherman from the 12th century perceives the world will vary significantly from a contemporary Italian animal ethics researcher’s perspective. The meaning of ‘fish’ for them differs. To the former, comprehending what a fish is involves understanding the practice of fishing. The fish might be viewed as a source of sustenance or as a sea dweller. Conversely, for the Italian animal ethicist, a ‘fish’ might simply be a biological being, a subject-of-a-life deserving respect, or an ethically controversial source of protein. When we assert that their form of life differs, we imply that their outlook on the world diverges, and such disparity manifests in the contrasting ways these individuals engage with and have learned how to use the concept of ‘fish.’ By engaging in different language-games, the meaning of the term varies due to its different usage.

### 3.1. Against Reduction to Theory

In general terms, Wittgensteinians maintain that the source of moral judgements is more complex than the orthodox approach allows. Following the motto “I’ll teach you differences” [47], Wittgensteinians urge us to pay attention to the multitude of conceptualisations that shape and enrich our moral deliberations. Oskari Kuusela, for instance, states that we should resist the idea that there are some “distinctively moral words or concepts […] necessary for morality” [48] (p. 85). We need not turn to principles to understand the rightness or wrongness of an action. Instead, the rightness or wrongness of an action is already embedded in ordinary language and practical concepts. As already mentioned, in line with anti-theorists (e.g., [49] (pp. 176–179)), Wittgensteinians resist the idea that principles can provide an exhaustive account of the morally relevant reasons characterising a particular context. As Anne-Marie Christensen puts it,

these thinkers turn their attention towards the fine-tuned description of the many ways in which moral considerations arise in human life, pointing to the gap between the simplicity of the picture of moral thinking offered by moral theories and the startling complexity of considerations and influences drawn on in actual moral practice [48] (p. 177).

Nevertheless, it should be noted that theory and principles are not necessarily repudiated. Some authors lower the relevance of theory and principles in our life by claiming that they do not have ultimate authority over our decisions, even if they can be helpful in clarifying the issues at stake. Moral theories may not provide “the most important form of understanding of our moral life”, but they can still provide “forms of general orientation” [20] (p. 208). In other words, our actions are informed by prima facie justificatory reasons rather than a decisive justificatory reasons. These prima facie reasons count in favour of or against any given moral judgement but cannot provide us with a categorical answer. Principles would then function as prima facie reasons that provide us guidance in seeing what is at stake, but cannot provide us with an ultimate solution with certainty.

### 3.2. Against Reduction of Animals to Animals Per Se

Wittgensteinians’ focus on the complexities of moral practices highlights a second way in which the orthodox account deflects us from our moral lives. Viewing animals as sentient beings oversimplifies the complexity of how we encounter animals. Orthodox approaches (specifically, utilitarianism) do not explore questions like “what does it mean to be a pet owner?” or “what kind of special relationship emerges between pet and pet owner?” Instead, they dissolve these differences, which arguably enrich our diverse experiences. The pest we want to get rid of, the wild animals we admire, and the pets we cuddle are substituted for an animal per se. But, we cannot and do not encounter animals per se in our forms of life. Lamenting this impoverishment of our moral life, Diamond writes:

It is a mark of the shallowness of these discussions […] that the only tool used in them to explain what differences in treatment are justified is the appeal to the capacities of the beings in question [27] (p. 468).

Indeed, framing oneself as a “pet-owner” rather than a “livestock-owner” comes with significantly different connotations, connotations that the notion of being a “sentient-being owner” does not seem to capture.

The reduction of a pet to a sentient-being is relevant for this discussion because it comes with an impoverished understanding of meaning and normativity. The concept ‘pet’ has an irreducible meaning for the relationships shaped through practices and language games specific to our forms of life. We learn what a pet is within special circumstances. This learning is built on playful and loving practices similar to those we enjoy with family members. We allow them in our house, we take care of them, and we show them love and affection [27] (p. 469).

These intricate networks of practices implicitly inform us how we ought to behave. They hold deep significance for us because they constitute part of our form of life and our understanding of ourselves as human beings [27] (p. 470). The reasons why it is wrong to eat one’s pet cannot be encompassed by the judgement that eating a sentient being is wrong. In fact, utilitarians cannot a priori exclude that we should eat our pets in certain circumstances. There will be circumstances where it is actually morally required by the principle of maximising overall happiness. If one’s pet dog were struck by lightning but still edible, then utilitarians cannot dismiss the idea that we ought to eat it (if doing so would maximise happiness) [26] (p. 374). The Wittgensteinian would argue that there is something deeply disturbing about eating one’s pet, something that is absent when eating a farmed animal. Both might be wrong, but the former is worse. There might be situations where the Wittgensteinians would concede that eating one’s pet is a reasonable thing to do but that would not make it a good action.

For Wittgensteinians, eating one’s pet can be thought of as an attack on the way we make sense of the world. This is why “if we call an animal that we are fattening for the table a pet, we are”, at best, “making a crude joke” [27] (p. 470). If we get rid of the concept ‘pet’, then we lose a number of justificatory reasons, reasons that utilitarianism cannot account for. Reasons that matter deeply to the agents involved in the situation and shape the agents’ moral understandings of the world.

### 3.3. Wittgensteinians’ Descriptive Proclivity

It is possible to characterise Wittgensteinian ethics as descriptive owing to its anti-reductionism [20,49,50]. We cannot reduce our moral understanding to a principle and we cannot reduce a pet to an animal per se. We are, then, confronted with a plurality of moral concepts and moral reasons. There is no ultimate guide for decision-making and action.

Wittgenstein claimed that “[p]hilosophy may in no way interfere with the actual use of language” [51] (p. 55); “it can in the end only describe it” [51] (p. 55). Ethicists can dedicate themselves to “arranging what we have always known” [51] (p. 52). They can do this by exploring how what already matters can be applied in different situations. They might show us that the animals we eat cannot be reduced to livestock or food, because they could be also meaningfully described as companions or wild animals to which we owe some reverence and respect. The ethicist’s job becomes that of someone who helps others to understand and reshape the world they perceive by making explicit the way they conceptualise the animal.

In sum, anti-reductionism and descriptivism motivate Wittgensteinians to reject the orthodox move of conferring normativity to reasons by means of principles. They understand our form of life as being the entwinement of practices, language, and meaning. This entwinement constitutes the ground of meaningful ethical discourse. It provides a theoretical framework that demands that we engage in a description of how we already conceptualise the world. These conceptualisations constitute the grounds for justification. They provide us with a much richer understanding of morality than a reduction to principle can allow.

As mentioned, Wittgensteinians might claim as follows: “A pet is not something to eat!” For Wittgensteinians, this claim itself provides a justificatory reason for why eating pets is wrong as it reminds us of a fundamental way in which we understand our relationship with pets. The irreducibility of the many reasons stemming from our forms of life is why “we do not eat our pets” is not merely an explanatory reason. It is what warrants treating descriptions as normative.

In the next section, we aim to show how social scientists can substantiate Wittgensteinian moral inquiry. In other words, we suggest what sort of empirical data Wittgensteinians can use to ground their moral judgments. We also argue that Wittgensteinians should do this if they wish to cogently address important moral issues.

## 4. Empirical Data for Ethical Guidance

In this section, we discuss how the social sciences can contribute to the Wittgensteinian approach to animal ethics. We provide a very general overview, which should help to ground an initial understanding of the issue. It is by no means exhaustive. Indeed, social sciences like sociology, anthropology, and psychology are diverse, varying in methodologies, fields of inquiry, and theoretical frameworks. We can nevertheless recognise the type of data that is valuable in this context.

To make our case, we draw on insights in Persson and Shaw’s paper “Empirical Methods in Animal Ethics” [18] which already discussed how empirical data from both natural and social sciences can inform orthodox approaches to animal ethics. Their classification regarding the significance of social science data is of particular interest as they identify six ways in which empirical data can contribute to ethical decision-making [18] (pp. 858–859). We shall focus on the most argumentatively salient three (Indeed, these authors claimed as well that empirical data from social sciences can be used for enhancing the contextual relevance and realism of ethics, describing factual information relevant to normative arguments and demonstrating the ethical dimensions of science, technologies, or organizations (Persson and Shaw 2015, 856). However as these are not strictly relevant for the present argument, we will not discuss them):Data about reasoning and moral beliefs held by agents involved in a specific practice [18] (pp. 858–859)Data about concrete examples of situations that can make “ethics more context-sensitive or realistic” [18] (pp. 858–859)Data about moral issues specific to a practice that has been overlooked by extant moral theories [18] (pp. 858–859)

Regarding 1, Persson and Shaw stress the importance of knowing about people’s opinions and beliefs: “without the knowledge of people’s awareness of the problem, there is no starting point for rules or guidelines” [18] (p. 857). Things like principles might be “too abstract” [18] (p. 858). It is, then, “unlikely that people are willing and able to stick to them in their context-dependent everyday life decisions” [18] (p. 858).

Regarding 2 and 3, Persson and Shaw argue that including such data in ethical discussions enables animal ethicists to ask more refined and precise questions. Persson and Shaw highlight the shift from broad inquiries like “does it matter whether their suffering can be compared to ours?” [18] (p. 858) to more focused considerations like ”is it morally acceptable what we do in line with our norms/laws to those creatures which certainly do suffer?” [18] (p. 858). They also point out that “qualitative data about people’s opinions could […] point to potential ethical issues” when it comes to aspects of human-animal interaction that have yet to be considered [18] (p. 857). Interviewing individuals engaged in specific moral contexts might uncover issues that moral theorists have overlooked. Such interviews could, for example, reveal conflicts between wildlife and humans, or conflicts within animal communities that ethical theorists have not anticipated.

Persson and Shaw’s claims are sound and in line with the views of a growing number of philosophers who suggest that:We should adjust moral theory to make it more realistic [17].It does not make sense to develop a practical philosophy that ignores our psychological dispositions [52].

If we want to implement norms, then we must seriously consider these two suggestions. However, proponents of such views are largely still aligned with the orthodox approach to animal ethics. They establish a set of ethical principles that tell us what is the right thing to do. They then proceed to try to find the most effective way to implement those principles [16] (pp. 176–181). In other words, they start from established moral principles, rather than from our forms of life, to secure what is morally relevant. They might claim that “there is the danger of moral relativism if context-sensitivity is overrated” [18] (p. 858).

Wittgensteinians would respond that context sensitivity cannot be overrated. It is the very ground that makes it possible to establish sensible moral discourse. Only concepts and norms that are already present at the moment of the inquiry constitute justifying reasons in moral discourse since the acquisition of new concepts takes time and involves the prolonged engagement of an agent in a practice. Ethical judgments should be grounded in the concepts that agents adopt, concepts like ‘pet’, ‘companion’, and ‘patient’ etc. This involves delving into the normative structure of forms of life (e.g., the practice of keeping pets). This how the social sciences can grant Wittgensteinians a more comprehensive and rigorous understanding of their topic. In fact, we contend that—given their approach to morality—Wittgensteinians can interpret these data as prescriptive, as affording justifying reasons.

## 5. Wittgensteinians and the Social Sciences

We might say that the social sciences can add meat to the bones of the animal per se. This can be done by describing what sorts of animals and what sort of norms can be found in any given context. Are we dealing with pets or patients? Is it wild or liminal life? And, what follows from this? The social sciences might provide the justifying reasons that Wittgensteinians claim the orthodox approach precludes. Qualitative data from a particular context would provide the Wittgensteinians the very concepts upon which to develop their ethical reflections. For example, social scientists might show us that animals are seen, e.g., as companions or as patients in certain contexts. These animals are treated in a way that the notion of ‘sentient being’ cannot properly describe. Indeed, every kind of practice prompts the need to gather data ad hoc, as Wittgensteinian hold that the meaning of concepts and words varies with their use, and therefore, with the different practices.

As mentioned, Wittgensteinians stress the role that context-dependent conceptualisations play in our moral life. They have also provided examples of which sort of concepts underpin our moral comprehension of the world. These might be concepts like ‘humanity’ [24,25,27,29,53], ‘pet’ [27,30], or ‘fellow creature’ [27,30]. However, when it comes to animal ethics, Wittgensteinians have not reached their conclusions through systematic and empirically informed analysis. They claim that concepts like ‘humanity’, ‘pet’, and ‘fellow creature’ help us understand aspects of our moral life. Yet, they remain ambiguous about (1) the extent of the consensus surrounding these concepts, and (2) the varying interpretations of them. That is, they did not show who holds these concepts, in which contexts they are employed nor whether they are understood in the same way by everyone.

Mostly, Wittgensteinians rely on literature and poetry to paint an imaginative (albeit rich) understanding of the relevant concepts. Diamond, for example, uses Charles Dickens’ *A Christmas Carol* to illustrate how we understand the concept of being human. Although we might accept that poetry and literature can provide a nuanced understanding of morality (many have argued for this, e.g., [32,54]) we contend that this is not sufficient to ground substantial Wittgensteinian ethical reflection. Indeed, the concepts that an author explores might not appropriately describe everyone’s understanding. Nor are they necessarily appropriate for every group of agents in different contexts.

For example, Wittgensteinians themselves do not share the same understanding of ‘pet’. Diamond argues that we understand what a ‘pet’ is because we let them into our houses [27]. Instead, when discussing the difference between our relationship with people with cognitive disabilities and our relationship with pets, Taylor, another Wittgensteinian, claims that

we clothe them but not our pets, seat them at the dinner table next to us and not place a bowl on the floor for them, they live in the house with us, not in a kind of kennel in the yard [30] (pp. 224–225).

Indeed, some people believe that the definition of ‘pet’ does not entail their residing indoors. For instance, besides those who confine their dogs and cats mainly to the outdoors, there are people who keep horses or chickens as pets, whose presence in houses and flats designed for human residence would cause significant trouble.

We think that the lack of a systematic inquiry into moral concepts is a significant shortcoming of current Wittgensteinian reflections on animal ethics. Moral concepts undergo changes in meaning and context over time. Philosophers cannot possibly possess a priori knowledge of such evolving concepts. They should rather rely on qualitative data to explore the conceptualisations people live by in the particular contexts in question.

The following two examples illustrate our point.

Firstly, the understanding of ‘pet’ can vary. If Wittgensteinians are interested in developing ethical reflections grounded in what the concept ‘pet’ means for a particular group of people, they should gather qualitative data about how the concept is used by that group of people. For example, Alison Sealey and Nickie Charles, following the idea that people’s experience and practices involving animals are reflected in their language [43] (pp. 486–487), [55], gathered qualitative data through open questions among UK citizens about the way they conceive animals. By asking, for example, ‘what is it that distinguishes a pet from other animals?’, they received a variety of different answers which show that within large populations, such concepts tend to vary [43] (p. 493), cf. also [56]. Generally, however, the understanding of ‘pet’ is often mixed with that of ‘family member’ [57,58,59]. There is empirical evidence that pets in Israel are increasingly considered to be part of the family. In fact, they can be explicitly assigned the status of ‘children’ (or implicitly assigned the status of ‘flexible persons’ [60] (p. 422). Data gathered in the US show that pets assume a crucial function in social [61] and familiar interactions [57,59] since they are sometimes described as the ‘glue’ that keeps the family together, or the beings around which one’s ‘household revolves’ [59] (p. 483). This does not mean that there are no other contexts where the same conditions apply. It is, though, an empirical question, one that should be tested in each different situation.

Secondly, the idea of applying a given concept to a novel situation may be inconceivable to some ethicists. They might need empirical evidence to stretch the limits of their imagination. Diamond initially doubted that the concept of ‘friendship’ could be extended to encompass all animals. She has, though, later on acknowledged that her perspective was limited. This was after she witnessed bondings between whales and Greenpeace rescuers [9] (p. 12) [27] (p. 470). Another example comes from the experience of Hannah Winther. She discusses the value that empirical inquiry can have when it comes to grasping the conceptualisations that different forms of life have about animals. She recounts how her ways of conceptualising salmon was enriched by learning how people from Norwegian and Samí cultures employ concepts like ‘iconic’, ‘beautiful’, and ‘amazing’ to describe salmon [3] (p. 11).

The kind of data from social sciences that should be employed by Wittgensteinians should then make explicit the way agents conceptualise animals in a given practice. These data can be gathered through qualitative interviews where questions such as “what is it that distinguishes a pet/a vermin/patient from other animals?” [43] (p. 493) can be asked to agents involved in a practice. A limitation of the present paper is that it cannot provide concrete and detailed examples of the kind of empirically derived qualitative data from which Wittgensteinian ethical reflection would most benefit. Indeed, we would need a data set about a particular context and about a particular problem to start with. For example, about the ethical issues concerning euthanasia in veterinary clinical practice in Austria. However, while we maintain that every ethical reflection should begin with an analysis of the concepts people engage with in a specific context, we would like to broadly outline how this process would look in general terms.

Let’s consider the moral complexities that veterinarians face when dealing with the euthanasia of, for example, a dog. Here the Wittgensteinian would be required to understand how the dog is conceptualised by the various agents. Beyond merely interacting with a sentient being, veterinarians grapple with diverse conceptualizations of the animal, influenced by how they and their clients perceive the dog. An inquiry into such perception would provide the Wittgensteinian with a clear idea of how the animal is conceived by different actors in the context of the clinic. Social sciences could then provide a systematically developed and transparent starting point (as opposed to the experiences that Diamond and Taylor have with their pets). As previously mentioned, pets might not only be seen as ‘family members’, but also as companions with whom people share emotional bonds and confide in [59,62]. Some even refer to them as “‘live-in’ therapists” [59] (p. 485) [63], highlighting their indispensable role in their owners’ lives. As a result, when veterinarians attend to a pet, they are not only dealing with a ‘patient’ but also with a being intertwined with specific practices that constitute the essence of the client’s life. The loss of a pet can lead to significant grieving for the owner [64] and this underscores the intricate nature of the veterinarian’s decisions, which cannot be simply reduced to maximizing the welfare of a being with interests. In certain situations, clinicians may even need to “encourage families to create healing rituals to mourn their loss and honor their companion animals” [59] (p. 490) cf. [65,66].

Exploring such conceptualisations allows the Wittgensteinian to gain insight about the differences between cases of euthanasia in small animal practice and cases of euthanasia of farm animals. Farmers might see animals as commodities or sources of income [67] (p. 9) rather than as ‘family members’. Concepts such as ‘commodity’ would highlight a tension between what the animal means for the farmer’s life with what it means for veterinarians who might find themselves as “advocates for the animal patient” [68] (p. 7). Horses, for example, might be seen in completely different ways whether they are kept as a pet or as livestock. It is by departing from such conceptualisations that the Wittgensteinian philosopher can develop their ethical reflections. However, what these conceptualisations are, must be explored on a case-by-case basis, following the Wittgensteinian idea that meaning is determined by use.

The above considerations provide important insight into the work of social scientists. The data collected by social scientists in interviews, focus groups, and questionnaires, are normatively relevant. They grant us a glimpse into the conceptual world that constitutes the moral landscape of a particular group of agents. Given how their anti-reductionist and descriptive perspectives are rooted in forms of life, Wittgensteinians are well advised to collaborate with social scientists. This can be fruitful when it comes to describing and evaluating moral practices in real-life complexities. It can, then, serve as a solid foundation for grounding Wittgensteinians’ moral criticisms. They should not wait and hope for the poet’s glimpse into the depths of new concepts. Rather, they should systematically and empirically explore how different understandings of concepts shape people’s understanding of moral problems.

## 6. Discussion

We have argued that Wittgensteinians should substantiate their ethical judgments with empirical data from the social sciences. An interesting question is whether social scientists are ethicists, given that they provide us with normative material. We think that the answer is “no”. Social scientists can describe how people think about moral issues, but they are not typically reflecting on how people think in order to judge whether they are right or wrong in doing a certain action. Social scientists can help us answer questions like “how are animals thought of in this or that practice?” Ethicists, whether orthodox or not, in contrast, ask questions like “is this way of conceptualising about animals appropriate?”, “are there other ways we can think about this problem?”, and so on. Ethics—at least the Wittgensteinian understanding of it that we have outlined in Section 3 is the activity or discipline of reflecting on already-held beliefs and already-established concepts. We are doing the work of ethicists as soon as we (1) reflect on the concepts we live by and (2) use those concepts as justificatory reasons (rather than as mere explanatory reasons).

This leads to a further question. If the line between descriptive and normative issues becomes indistinct—if the line between ‘is’ and ‘ought’ becomes blurred—would this justify the status quo? If so, then Wittgensteinian approaches offer little in comparison to orthodox approaches, as they would be defending an ethical perspective incapable of providing any moral criticism. One might even be led to question the purpose of this article. Why propose improving an animal ethics position that is ultimately not worth considering? This is not the primary focus of our discussion; addressing it in detail would require more than is possible in this paper. It is, nonetheless, necessary to provide a brief response to such concerns.

Advocates of the orthodox approach are sometimes concerned that Wittgensteinians would defend the status quo, because they are simply “reducing ethics into sociology” [69] (p. 37). By surrendering the high ground that moral principles offer, the ethicist can no longer criticise a given state of affairs. This concern relies on a misunderstanding of the Wittgensteinian position. It is true that Wittgensteinians claim that criticism must rely on forms of life. However, this does not mean that we should accept any “of the multitudinous contents that happen to be subsumed by that form” [70] (p. 332). As a matter of fact, though Wittgensteinians tend to refuse that there is some absolute point from which we can judge everything, they still concede that there are concepts, or as Diamond calls them “guides to thinking” that survive the test of time [71] (p. 303). So even if they develop, so to speak, a bottom-up approach, this does not mean that various forms of life do not share part of the same ground. Wittgensteinians tend to emphasize anti-reductionism and the need to accept that every “exploration has to be carried out from a position of immanence within linguistic practice” [72] (pp. 97–98). But this in no way entails that there is *one* linguistic practice which is the right one.

Discussing the ethics of animal experimentation, Diamond remarks that a “laboratory rat is neither a machine nor a person; if it were one or the other, there would be no problem with how to draw the boundaries of morality” [26] (p. 346). How we should conceptualise the rat is undecided (let alone how we ought to treat it). What is clear is that there is something off—something morally distasteful—about treating an animal as a research tool. The problem arises from within a form of life. There is no need to appeal to a principle of utility maximisation. Wittgensteinians contend that the variety of ways in which we conceptualise animals in our forms of life provides sufficient resources for ethical critique and change [20,38].

This means that the Wittgensteinian approach does not merely provide us with (1) a description of the conceptualisations that constitute an understanding of a particular practice, and (2) the nuances that characterise a context. Rather, it helps us to reflect on these descriptions and nuances. It asks us to try out different concepts and compare them against each other [26]. Is this animal necessarily a pest? Or can we treat it like a liminal noisy neighbour? Is the laboratory rat a tool? Or, does it make, after all, sense to conceive it as a companion?

The only move that Wittgensteinians refuses to make is to reduce the moral landscape to a set of principles, a set of principles that putatively grants us a high ground from where to judge human-animal interactions. For Wittgensteinians, we never have and never will attain such a high ground. We might say that our moral landscape is full of hills and valleys. It is a landscape that we must explore when confronted with a moral problem that requires workable solutions. Within the plurality of moral concepts that constitute forms of life, we must reflect on—we must take account of—every extant element. There is no ultimate conceptualisation that we can ‘force’ on reality and then declare “this is how the world is!” cf. [73].

Wittgensteinianism cannot be hasty dismissed. The Wittgensteinian approach might allow us a methodologically fine-grained understanding of various situations. It can provide us with the tools for a nuanced moral criticism of the ways we deal with animals. Different communities are entrenched in different webs of meaning, different practices, and different uses of languages. To successfully explore the plurality of conceptualisations that constitute such webs, Wittgensteinians are well advised to collaborate with social scientists. With these resources, Wittgensteinians might demonstrate how their account offers a more fine-tuned understanding than the one offered by orthodox accounts.

## 7. Conclusions

We have argued that incorporating empirical data from the social sciences is essential for Wittgensteinian approaches to animal ethics because such data can determine Wittgensteinians’ ethical reflections by identifying the concepts upon which the criticism can start. Wittgensteinian approaches require empirical data to comprehend the foundations on which their criticism of orthodox approaches is based. Given their anti-reductionist and descriptive stance, Wittgensteinians identify a broad range of elements as justificatory reasons. These elements, they claim, characterise our lived morality in a way that orthodox approaches do not. Wittgensteinians argue that the reasons why we should hold a particular moral judgment are found in the concepts that constitute our understanding of morality within specific forms of life. For example, the wrongness of eating one’s pet exceeds the violation of a principle that prohibits us from harming a sentient being. Instead, it reflects an action that disrupts our form of life and thereby threatens the meaning of a good life.

Wittgensteinian approaches to animal ethics have mostly refrained from systematically engaging in empirical analyses during ethical reflection. They have, instead, traditionally turned to poetry and literary fiction to exemplify the richness of moral life. Although we acknowledge the importance of literary works in enhancing moral understanding, we contend that the social sciences can offer valuable information. This is because concepts evolve over time and can vary across different forms of life.

In short, Wittgensteinian ethics needs empirical data from the social sciences if it is to live up to its own methodological promises of providing us a more fine-grained understanding of ethical problems.

## Data Availability

Not applicable.

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
