# Peer review of "What Is Wrong with Eating Pets? Wittgensteinian Animal Ethics and Its Need for Empirical Data"

_animals, 2023, doi:10.3390/ani13172747_

Round 1

Reviewer 1 Report

  • The paper is extremely interesting. However, it does not meet the MDPI requirements (I.e., there is no experimental design, results are not reproducible). Perhaps it might be submitted to another Journal.

Author Response

Dear reviewer,

Thank you very much for taking the time to read our manuscript and for the positive feedback!

We submitted the paper because we thought it fits the special issue “Empirical Animal and Veterinary Medical Ethics” well. The special issues areas of interest are:

1) Reflections on the foundation, meaning or possible scope of empirical animal and veterinary medical ethics;

and

2) Guidance and discussion concerning methodologies of empirical animal and veterinary medical ethics

Best regards

Reviewer 2 Report

The topic is appropriate for Animals, and the thesis is plausible and significant. Wittgensteinian ethical analysis appeal to forms of life, the best evidence for and understanding of forms of life are provided by empirical investigations of social science, and therefore Wittgensteinian ethical analyses would be improved by incorporating and being in conversation with such social science research. I think there are 3 main barriers to publishability at this point. First, the characterization of “Wittgensteinian ethical analyses” in contrast to orthodox approaches is unclear and confusing. Second, the argument would be substantially strengthened if more examples of the kind of social science work that the authors think should be included in Wittgensteinian analyses along with detailed discussion that illustrates how such inclusion would improve the Wittgensteinian analyses. Third, there are several typos and grammatical mistakes that need to be fixed prior to publication. My overall recommendation is “Reconsider after major revision)”.

Lines 56-72: organization of the paragraph. The key feature of an orthodox approach appears to be “we first identify the correct theory (with associated principles) and only then do we apply it in the most effective way”. But this explanation occurs (lines 67-68) only after several sentences discussing the orthodox approach. It would be helpful for the reader to have the defining feature of the orthodox approach introduced immediately after the term is first used (lines 57-58).

The author’s “conclude that a systematic inquiry into how people actually think about animals can provide the foundation for sound Wittgensteinian ethical reflection” (lines 118-120) but earlier said they were not going to “settle the question of whether Wittgensteinian approaches do indeed provide a better understanding of moral problems than orthodox approaches: (lines 92-94) and subsequently say that “our paper is not aimed at defending a Wittgensteinian position” (lines 125-126). But if the conclusion is that the Wittgensteinian reflection is, indeed, “sound,” then that at least suggests that it has been defended as the correct method and hence, better than orthodox approaches. This tension in describing how ambitious the aims of the paper are should be resolved.

Lines 138-139: the reasoning here should also include a comparison of the harms to the animal and the benefits derived from eating, as well as a comparison to alternatives available.

Lines 155-157: since the Wittgensteinian critique is supposed to apply to deontological approaches as well, it is important to provide more detail in how the deontological approach works here and why it is similarly criticizable. After all, deontologists typically take themselves to be giving necessary conditions for moral permissibility (i.e., it is unethical to treat an individual with moral status merely as a means; it is unethical to harm an individual with moral status absent meeting some strong threshold). But such principles can be consistently supplemented with additional principles such as: it is unethical to abandon one’s pet; it is unethical to kill one’s pet for trivial reasons, etc.) These may well make essential reference to the concept of a pet.

Lines 175-178: This Wittgensteinian remark doesn’t answer the question. Even someone who firmly believes without doubt that one should not eat pets and would never dream of doing it, can still ask questions about what makes it wrong. Merely reiterating “you should not eat pets” doesn’t answer that question. An if we understand “a pet is not something to eat” as meaning that pets belong to a category such that it is wrong to eat things in that category, that also doesn’t answer the question of what’s wrong with eating things in that category. So it isn’t clear what alternative explanation is being proposed by the Wittgensteinian as opposed to the explanation provided by proponent of one of the orthodox approaches. What exactly is the disagreement here? Is it a disagreement about whether a request for reasons is appropriate? It would seem like it is open to the orthodox approach folks to agree that a request for such reasons might be inappropriate (as demonstrating extreme moral ignorance, or conversationally implying that one doubts that it is wrong to eat pets, or something like that). Or is the disagreement about which reasons explain why you should not eat pets? If so, then the example might work better if you phrased it as a response to the question: “Why is it wrong to eat Fido?” (about a specific example) and the Wittgensteinian responds by saying “Because Fido is a pet” whereas the orthodox approaches would respond in different ways. But, isn’t “one should not eat pets” itself a “principle that informs us about the right ways to act,” the requirement for which is supposed to be part of what sets the orthodox approaches apart from the Wittgensteinian approach?

Line 195: What is the referent of “this” which is “precisely what Wittgensteinians dispute”? The orthodoxer has just claimed that the Wittgensteinian does not rely on a normative principle, but that claim appears to be accurate and not in dispute, given the authors’ description of the Wittgensteinian as someone who doesn’t ground judgements in principles but in forms of life (lines 205-206).

Line 213-214: I don’t know any view, Wittgensteinian or otherwise, that would try to reduce the concept of ‘pet” or the practice of pet-keeping to a moral prohibition; that just seems like a category mistake. Practices and concepts aren’t prohibitions. Can this be rephrased to state more clearly the point the authors are trying to make? Perhaps something about the moral significance of something’s being a pet.

Lines 230-242: This description of the Wittgensteinian approach would benefit from walking through an illustrative example or two. What is an clear, easy-to-understand example of “a form of life”. What is a clear, easy-to-understand example illustrating how “meaning is constructed within a community? What is a clear, easy-to-understand example showing how a moral judgment is “shaped within” a language game? What does it even mean to say that that morality is constituted in a form of life? Without clear examples, these bits of jargon will be obfuscating and confusing to people not already familiar with Wittgenstein.

Lines 300-302: for the (standard) utilitarian, the only things that are intrinsically good or bad are pain and pleasure. Eating one’s pet is neither of those things. And the (standard) utilitarian begins with an assessment of how much good and bad there will be in the world if the agent performs any of the various options open to them, and then draws conclusions about what is morally required. They do not, as portrayed here, start with what is morally required, and then conclude that that must be good. (in slogan form, according to the Utilitarian, the good is prior to the right.)

Lines 397-398: Why can’t the agent acquire new concepts that would then constitute justifying reasons? Unless the concepts in question are innate and present from birth (?) that presumably is exactly what happened at some point.

Line 413: notions can’t grasp. Perhaps “explain” instead of “grasp”.

Lines 429-471: A central claim in the authors’ argument is that Wittgensteinian ethical analyses would greatly benefit from being informed by social science investigations of the forms of life that the Wittgensteinian ethicists are appealing to. Key examples to support this claim are discussed at lines 429-471. But of the people discussed here, Hannah Winther, Herwig Grimm, Susana Monsó, Cora Diamond, Craig Taylor, Cheryl Abbate, are all trained as philosophers and not trained as social scientists and only Shir-Vertesh is trained as a social scientist (an anthropologist). If anything, this supports the opposite of the conclusion being argued for. What is needed here are (hopefully numerous) examples of the kinds of social science work the authors believe should be incorporated into Wittgensteinian ethical analyses along with a detailed discussion of how that work would improve and enhance the analyses.

Line 55: “conducing” should be “conducting”

Lines 72: “account” should be “accounts”

Lines 80-81. Sentence is garbled.

Line 222: “withing” should be “within”

Line 227: the sentence is garbled.

Line 427-429: sentence is broken into two incomplete sentences.

Line 507: “Wittgenstein,” should be “Wittgensteinians”

Author Response

Dear reviewer,

We would like to thank you for taking the time to review our manuscript and providing such detailed and helpful comments!

Please see the attachment for our detailed answer,

Best regards,

Reviewer 3 Report

An excellent paper bringing a new perspective to the issue of animal ethics. The inclusion of inputs from various social sciences are common in animal welfare science literature, and should form an important part of ethical enquiry too. 

As predominantly an animal welfare scientist, with an interest in ethics, rather than visa versa, my approach to the matter is more 'Karl Popper', than 'Ludwig Wittgenstein', but the paper makes a new, interesting and useful contributions to the debate, and which should be published. The logic in the arguments is valid, even if I have some  personal reservations about the premise on which they are built; however such debate is the meat on which advances in ethics are made. Thank you for an interesting read.

Author Response

Dear reviewer,

Thank you for taking the time to read the paper and for your positive feedback.

 We are glad to hear that you found the paper's contribution useful and appreciate the value of our work!

Best regards,

Round 2

Reviewer 1 Report

To this end I don't have any other elements on hand to make additional comments to my first revision.

Author Response

Dear Reviewer,

Thank you again for taking the time to read our manuscript and for the positive feedback!

Best regards

Reviewer 2 Report

The paper is much improved. Although some parts are still a bit murky, I think some of that is inherent to exploring Wittgensteinian ethical approaches and the authors have done as good a job as can be reasonably expected. The argument is clearer and stronger with respect to its critique of utilitarian approaches than it is to its critique of deontological approaches, which are more varied and open-ended, but I think they still make good points. In sum, my major concerns have been addressed

Lines 107-108: The authors could add that Wittgensteinians (e.g., Diamond and Taylor) have drawn on their own individual, anecdotal preconceptions as well.

Line 502-503: The authors might also mention that not everyone views their pets as things that live, literally, in their house. Horses, for example, some kinds of dogs, chickens, etc.

Some minor edits are still needed, but nothing that interferes with the paper's overall readability.

Author Response

Dear Reviewer,

Thank you again for taking the time to review our manuscript and for the helpful feedback.

We have considered your comments and modified our manuscript accordingly.

Please see the attachment for our detailed answer.

Best regards,
